# Identification of Novel Genetic Variants in a Cohort of Congenital Hypogonadotropic Hypogonadism: Computational Analysis of Pathogenicity Predictions

**DOI:** 10.3390/ijms26115207

**Published:** 2025-05-28

**Authors:** Paola Chiarello, Gianmarco Gualtieri, Sabrina Bossio, Giuseppe Seminara, Marianna Molinaro, Gemma Antonucci, Anna Perri, Valentina Rocca, Rossella Cannarella, Sandro La Vignera, Aldo E. Calogero, Emanuela A. Greco, Rodolfo Iuliano, Stefano Alcaro, Antonio Aversa

**Affiliations:** 1Department of Pediatrics, Dulbecco Azienda Ospedaliero-Universitaria of Catanzaro, 88100 Catanzaro, Italy; paola.chiarello@unicz.it; 2Department of Experimental and Clinical Medicine, Università degli Studi Magna Græcia di Catanzaro, 88100 Catanzaro, Italy; sabrina.bossio@unicz.it (S.B.); dott.giuseppeseminara@gmail.com (G.S.); marianna.molinaro@studenti.unicz.it (M.M.); anna.perri@unicz.it (A.P.); valentina.rocca@unicz.it (V.R.); 3Dipartimento di Scienze della Salute, Università degli Studi Magna Græcia di Catanzaro, Campus “S. Venuta”, Viale Europa, 88100 Catanzaro, Italy; g.gualtieri@unicz.it (G.G.); iuliano@unicz.it (R.I.); 4Department of Pharmacy, Health and Nutritional Sciences, University of Calabria, 87036 Rende, Italy; gemma.antonucci@unical.it; 5Department of Clinical and Experimental Medicine, University of Catania, 95123 Catania, Italy; rossella.cannarella@unict.it (R.C.); sandrolavignera@unict.it (S.L.V.); acaloger@unict.it (A.E.C.); 6Glickman Urological & Kidney Institute, Cleveland Clinic, Cleveland, OH 44195, USA; 7Dipartimento di Scienze Economiche, Psicologiche, della Comunicazione, della Formazione e Motorie, Nicolò Cusano University, 00166 Rome, Italy; emanuela.greco@unicusano.it; 8Net4Science srl, Università degli Studi Magna Græcia di Catanzaro, Campus “S. Venuta”, Viale Europa, 88100 Catanzaro, Italy

**Keywords:** CHH, KS, nIHH, genetic variants, VUS, computational analysis

## Abstract

Congenital hypogonadotropic hypogonadism (CHH) is a rare and heterogeneous genetic disorder with variable penetrance caused by GnRH deficiency, leading to delayed puberty and infertility. In 50–60% of cases, CHH is associated with non-reproductive abnormalities, most commonly anosmia/hyposmia (Kallmann syndrome, KS). Over 60 genes have been implicated in CHH pathogenesis. We aimed to perform genetic screening in a cohort of 14 patients (10 males, 4 females; mean age 22 ± 7.72 years) with suspected or diagnosed HH/KS. Genetic analysis was conducted using next-generation sequencing (NGS) with a custom panel of 46 candidate genes. Variant interpretation followed ACMG standards and guidelines. Multiple tools were used to predict the structural effects of variants on tertiary protein structure, assessing their pathogenicity. Novel variants were functionally characterized by qRT-PCR on mRNA extracted from peripheral leukocytes. NGS identified nine rare variants and four novel variants in genes previously associated with normosmic isolated HH (nHH) and/or KS (*FGFR1*, *PROK2*, *TAC3R*, *DCC*, *WDR11*, *IL17RD*, *DUSP6*, *KAL1*, *FGF8*, *IL17RD* and *DCC*). The variant in *TAC3R* (p.Trp275Ter) was pathogenic; variants in *ANOS1* (c.541+1G>A), *IL17RD* (c.1303_1304dup, p.Lys436ThrfsTer58), and *TAC3R* (p.Lys361Ter) were likely pathogenic. Nine variants were classified as variants of uncertain significance (VUS). Our study identified a possible genetic cause in 71% of the CHH/KS cohort, emphasizing the importance of genetic screening and functional characterization of genetic variants in patients with a phenotypically and genetically heterogeneous disorder like CHH.

## 1. Introduction

Congenital hypogonadotropic hypogonadism (CHH) is a rare disorder characterized by a deficiency in the production, secretion, or action of gonadotropin-releasing hormone (GnRH), leading to delayed puberty and infertility. GnRH is a key regulator of reproduction in both sexes. This neuropeptide is released by specific hypothalamic neurons in a pulsatile manner and binds to type 1 receptors on gonadotropic cells in the anterior pituitary. This stimulates the synthesis of luteinizing hormone (LH) and follicle-stimulating hormone (FSH), which, in turn, promote the production of sex steroid hormones by the gonads [1]. Biochemically, CHH is characterized by low or inappropriately normal levels of gonadotropins, with low estradiol levels in females and low testosterone levels in males.

Clinically, CHH in males is characterized by absent or minimal virilization, poor libido, and erectile dysfunction, with cryptorchidism and microphallus being typical in infant boys. Adult females with CHH typically present with primary amenorrhea, and both sexes may exhibit eunuchoid proportions. When anosmia or hyposmia is present, CHH is diagnosed as Kallmann syndrome (KS), which results from a disruption in the embryonic migration of GnRH neurons from the olfactory placode to the hypothalamic nasal region, where these neurons normally settle and differentiate permanently [2]. The prevalence of KS is estimated to be 1 in 40,000 in females and 1 in 8000 in males [3].

Both CHH and KS are phenotypically and genetically heterogeneous, with the risk of transmission primarily dependent on the specific gene mutation. These disorders can occur sporadically or in familial cases, and several modes of transmission have been identified, including autosomal recessive, autosomal dominant, X-linked inheritance, and oligogenic transmission models. Oligogenic transmission in CHH/KS has been increasingly demonstrated in the past decade [4].

The advent of next-generation sequencing (NGS) has significantly advanced the identification of new genes responsible for the diverse phenotypic expression of CHH/KS. However, while NGS has greatly contributed to diagnosis and management, whole-exome sequencing (WES) is limited compared to whole-genome sequencing (WGS), as WES only analyzes coding regions and misses intronic or regulatory regions, as well as large rearrangements, which may also play a critical role in genetic diseases [5]. Additionally, distinguishing true oligogenism from cases where multiple rare variants are identified without clear clinical impact is a growing challenge. In this context, advancements in in vitro and in silico approaches, alongside the development of computational algorithms, now enable more accurate assessment of the pathogenicity of identified variants, reducing the occurrence of variants being classified as variants of uncertain clinical significance (VUS) [5].

To date, more than 60 different causal genes have been identified in approximately 50% of CHH/KS cases: *KAL1* (*ANOS1*), which is X-linked; *FGFR1* (encoding fibroblast growth factor receptor 1), *FGF8*, *CHD7*, *HS6ST1* (encoding heparan-sulfate 6-O-sulphotransferase 1), *SOX10*, *SEMA3A* (encoding semaphorin-3A), *WDR11* (encoding WD repeat-containing protein 11), *IL17RD* (encoding interleukin-17 receptor D), which follow autosomal dominant inheritance, *PROKR2*, *PROK2*, and *FEZF1* with autosomal recessive inheritance. Additionally, genes such as *GNRHR* (encoding the GnRH receptor), *GNRH1* (encoding GnRH 1), *KISS1R*, *KISS1*, *TACR3*, and *TAC3* are involved in normosmic CHH (nCHH). Interestingly, *FGFR1* and *PROKR2* mutations can occur in both KS and nCHH patients [3].

In this study, we performed genetic screening using NGS on 14 patients with suspected or previously diagnosed CHH/KS, and through computational analysis and molecular biology approaches, we investigated the pathogenicity of the identified variants.

This study was conducted in accordance with the ethical principles for medical research involving human subjects set forth in the Declaration of Helsinki and was approved by the local Ethics Committee (CET Calabria, Catanzaro, Italy, prot. n. 88 28 November 2023). Written informed consent for publication was obtained from all the patients.

## 2. Results

### 2.1. Clinical Features of the HH Cohort

We enrolled 14 patients diagnosed with hypogonadotropic hypogonadism (HH) (10 males, 4 females, mean age 22 ± 7.72 years), 5 with nCHH, 7 with KS, and 2 with adult-onset HH, following the clinical, hormonal, and radiological criteria of the European Consensus Statement [6]. These patients exhibited several clinical and genetic characteristics (Table 1). Based on the smell test, patients were classified as normosmic (n = 6, 57%) and hypo/anosmic (n = 8, 43%). Hypoplasia/agenesis of olfactory bulbs was found in 7% of patients (1/7 hypo/anosmic patients). Remarkably, olfactory structures were normal in 6 anosmic patients, as documented by magnetic resonance imaging (MRI) studies. Of 14 patients, 2 (14%) presented neurosensorial hearing loss of various degrees (1 hypo/anosmic, 1 normosmic), as documented by functional testing. Renal ultrasound revealed 1 case with renal anomalies (anosmic). No midline defects were found, including abnormal palate, dental anomalies, pectus excavatum, bimanual synkinesis, iris coloboma, and absent nasal cartilage. Anamnestically, 1 anosmic patient reported cryptorchidism and micropenis. A total of 4 patients (44%), all anosmic, reported eunuchoidal proportions. Other clinical characteristics were ataxia that started 20 years later in a female patient with adult-onset HH and severe osteoporosis in one patient, affected from reverse nHH (Table 1).

### 2.2. Genetic Variants and Their Pathogenicity

Our study identified a genetic cause in 71% of CHH/KS patients.

At NGS analysis, several CHH probands harbored a total of 9 rare sequence variants and 4 novel sequence variants in genes previously identified to be associated with nHH and/or KS: *FGFR1* (n 1), *PROK2* (n 2), *TAC3R* (n 1), *DCC* (n 2), *WDR11* (n 1), *IL17RD* (n 2), *DUSP6* (n 1) and *KAL1* (n 1), *FGF8* (n 1), and *IL17RD* (n 1), respectively.

These variants included 1 pathogenic (rare), 3 likely pathogenic (2 unique and 1 already reported by other authors), and 9 of uncertain significance (VUS) (5 unique). Causal pathogenic and likely pathogenic variants are distributed across 3 genes: *ANOS1* (likely pathogenic), *IL17RD* (likely pathogenic), and *TAC3R* (pathogenic and likely pathogenic). The VUS identified affected 6 genes: *DCC* and *IL17RD* (3 patients each), *PROK2* (2 patients), *FGFR1*, *FGF8* and *WDR11*, and *DUSP6* (1 patient each). In one female CHH patient, we identified 2 rare benign sequence variants (0.7 and 0.002597 in gnomAD) in *CHD7* and *TACR3* genes. All are present in the heterozygous/hemizygous state (Table 2).

### 2.3. Oligogenic Transmission in Three Patients with CHH

Considering all pathogenic, likely pathogenic, and VUS, 3 patients (21%) exhibited variants in 2 genes; 2 female patients with VUS variants in *WDR11* and *DCC* and in *FGF8* and *IL17RD*, respectively; 1 female patient with pathogenic and likely pathogenic variants in *TACR3*. For VUS variants, the frequency of oligogenicity was 44%.

### 2.4. Frequency of Variants According to Phenotype

The frequency of causal pathogenic and likely pathogenic variants in patients with KS and nHH was 14% (1 out of 7) and 29% (2 out of 7), respectively. The frequency of VUS variants in patients with KS and nHH was 71% (5 out of 7) and 86% (6 out of 7), respectively. Only 1 patient with KS had 2 rare benign variants. VUS variants were identified in 4 out of 5 patients with eunuchoidal proportions (2 in *PROK2*, 1 in *FGFR11*, and 1 in *DUSP6*). In our cohort, pathogenic and likely pathogenic variants were not associated with non-reproductive phenotypes. In the 2 patients with clinical occurrence of cryptorchidism, we did not find mutations. A family history of CHH was reported in 1 patient with a likely pathogenic variant in *ANOS1*. A family history of pubertal delay was identified in 2 patients with VUS variants in *PROK2* and *IL17RD*.

### 2.5. Functional Characterization of ANOS1 and IL17RD Mutations

The new mutation described in patient 1, identified in *ANOS1* gene (c.541+1G>A), was classified as class 4, probably pathogenetic. It is a basic substitution in canonical splicing with probable loss of function and results in a premature stop codon. The variant was not previously reported in the reference population databases, including the Genome Aggregation Database (GNOMAD), Exome Aggregation Consortium (ExAC), ClinVar, and Human Gene Mutation Database (HGMD). His brother received a diagnosis of KS in adulthood, and he carried the same mutation. Accordingly, their mother was heterozygous for the mutation (Figure 1).

The new mutation described in patient 2 is a loss-of-function variant in the *IL17R* gene (c.1303_1304dup (p.Lys436ThrfsTer58)). This mutation is a frameshift, and it was classified as class 4, as probably pathogenetic (Figure 2). The variant is of maternal origin. qRT-PCR was performed to assess our probands’ mRNA expression levels of *ANOS1* and *IL17R*. The results clearly showed that c.541+1G>A and p.Lys436ThrfsTer58 mutations affect mRNA content; specifically, the level of *ANOS1* and *IL17RD* mRNA was significantly reduced in patient 1 and patient 2, respectively, compared to the control (Figure 3).

### 2.6. Molecular Dynamics Simulations of WDR11 Variant

The WD repeat-containing protein 11 (*WDR11*) is known for its typical WD40 domain, a seven-bladed β-propeller structure with a characteristic donut-like shape. Notably, the central pore of this domain plays a crucial role in facilitating interactions with peptide regions of key binding partners. Each blade of the WD40 domain typically contains a conserved glycine-histidine and tryptophan-aspartate (WD) motif. The β-propeller fold of these domains is structurally adaptable, allowing the domain to maintain its integrity despite deletion or insertion of WD repeats, which can range from 5 to 8 repeats [7]. At the start of the computational analysis, no experimental structures were available for *WDR11*. As a result, an AlphaFold-predicted model (AlphaFoldDB code AF-Q9BZH6) was used to isolate the protein domain containing the M769R mutation, retaining residues from Ala465 to Ala828 (Figure 4).

The selection of these residues was guided by visual inspection of the model. ALA465 was retained to maintain a minimal, flexible tail that has a limited effect on structural stability during MD simulations. Similarly, Ala828 was chosen as the endpoint because residues Met825 and Ser827 are crucial for stability. Met825 is oriented inward, while Ser827 forms a key interaction with Arg819. Removing these residues would likely compromise the stability of the entire domain. Consequently, due to the truncation, the amino acid numbering was reset, and the methionine previously labeled as position 769 is now renumbered as 306. At the start of the computational analysis, no experimental structures were available for *WDR11*. As a result, an AlphaFold-predicted model (AlphaFoldDB code AF-Q9BZH6) was used to isolate the protein domain containing the M769R mutation, retaining residues from Ala465 to Ala828 (Figure 4). We selected these residues by visual inspection of the model. The selection of these residues was guided by visual inspection of the model. Ala465 was retained to maintain a minimal, flexible tail that has a limited effect on structural stability during molecular dynamics simulations (MDs). Similarly, Ala828 was chosen as the endpoint because residues Met825 and Ser827 are crucial for stability. Met825 is oriented inward, while Ser827 forms a key interaction with Arg819. Removing these residues would likely compromise the stability of the entire domain. Consequently, due to the truncation, the amino acid numbering was reset, and the methionine previously labeled as position 769 is now renumbered as 306.

In August 2024, Deng et al. published the cryo-EM structure of the human WDR11-FAM91A1 complex, available in the Protein Data Bank under PDB codes 8Z9M and 8XFB [8]. Both cryo-EM structures are missing the segment spanning His622 to Ser674. As shown in Figure 5 to evaluate the accuracy of the AlphaFold (AF) model, we superimposed it onto the WD40-2 domain of the resolved cryo-EM structures and calculated the root mean square deviation (RMSD) between them (Figure 5). An RMSD below 1 Å demonstrates the high predictive accuracy of the AF model, affirming its suitability for our computational studies. Additionally, AlphaFold generates a per-residue confidence score, known as predicted local distance difference test (pLDDT), ranging from 0 to 100. Scores below 50 typically indicate that a region may be unstructured in isolation. In our selected model, the pLDDT remains above 90 for most of the structure, except for the missing segment, which shows pLDDT values ranging from 70 to 50 or lower than 50. The lower scores for this absent region are consistent with its expected high flexibility, further highlighting the challenges in experimentally resolving this portion of the structure.

MDs were conducted for 500 ns on both the wild-type and mutant M769R complexes (Figure 6). To assess dynamic conformational changes throughout the simulations and evaluate the mutation’s effect on overall protein stability, we calculated the C-alpha RMSD relative to the initial frame, aligning the trajectories based on a stable set of C-alpha atoms. As shown in Figure 6, both systems undergo an initial adjustment phase during the first 100 ns. After this period, the C-alpha atoms of the wt complex show greater stability compared to those of the M769R mutant. While the RMSD values for both systems generally remain within the 2–4 Å range, the M769R mutant exhibits notable fluctuations, with peaks reaching up to 5 Å at certain points in the simulation. This behavior in the M769R mutant suggests that it may not adopt a topology similar to the wt complex, potentially reflecting a destabilizing effect associated with the mutation.

Root mean square fluctuation (RMSF) analysis was conducted to examine the differences in local structural fluctuations between the wt and M769R systems. As illustrated in Figure 7, both wt and M769R display similar fluctuation profiles across most protein residues (Figure 7). However, around residue 150 and beyond, notable peaks indicate that the M769R mutant exhibits significantly higher fluctuations than the WT protein, particularly between residues 150 and 200, where RMSF values exceed 8 Å. This region, referred to as the “helix zone”, contains two short α-helices predicted by AlphaFold. Known for its high flexibility, this segment has been challenging to resolve in experimental structural studies. The observed increase in flexibility within the helix zone for the M769R mutant may be linked to the unique structural organization of the WD40 domain in *WDR11*. This domain consists of four-stranded β-sheets arranged in a circular, pseudosymmetric pattern, similar to propeller blades, which contributes to the compactness of the overall structure. The M769R mutation may disrupt this stability, potentially propagating structural changes to the helix zone. However, obtaining the actual structure of this region would be invaluable for confirming this hypothesis and providing a more detailed understanding of the underlying mechanisms.

To investigate the impact of the M769R mutation on intradomain motions within the protein, dynamic cross-correlation matrices (DCCM) were calculated for each residue in both configurations (Figure 8). Overall, the mutant M769R displays a shift in correlation patterns compared to the wt. Specifically, the mutant exhibits more extensive anti-correlated regions, indicating an increase in opposing motions between residue pairs. In terms of the helix zone’s self-correlation (residues 150 to 200, orange circle in Figure 8), the wt shows this region as a compact, highly correlated block, indicating coordinated motion within residues 150–200. In contrast, the mutant displayed weaker and more fragmented correlations within this zone, suggesting that the mutation introduces additional flexibility or instability. This observation is consistent with the higher fluctuations observed in the RMSF analysis for this region in the mutant. Regarding the impact of the mutation site’s motions on the helix zone (yellow circle in Figure 8), an anti-correlation is observed between residue 306 and the 150–200 region in the wt protein, indicating opposing motions between these areas. However, in the mutant, this relationship shifts to a positive correlation, suggesting that these regions now move more coordinately. This transition from anti-correlation to correlation indicates that the M769R mutation has altered the dynamic relationship between these regions, likely through new interactions or structural changes that align their motions. Such a shift could lead to a loss of native flexibility or stability, potentially affecting the protein’s overall function. In summary, the M769R mutation appears to induce a more complex and disordered dynamic profile, with expanded regions of both correlation and anti-correlation throughout the structure. This altered dynamic behavior is likely a key factor in changes to protein stability and functionality.

## 3. Discussion

We report 14 cases of CHH/KS linked to novel genetic mutations. Our study identified a potential genetic cause in 71% of patients, aligning with the prevalence observed in other CHH cohorts. Previous systematic reviews estimated the worldwide prevalence of causative genetic variants to be approximately 31% in KS and 23% in nHH [9,10].

We describe a patient with KS, exhibiting a novel hemizygous variant in the *ANOS1* gene (c.541+1G>A) and a patient with cHH carrying a novel loss-of-function variant in the in a deep intronic region of the *IL17R* gene (c.1303_1304dup (p.lys436ThrfsTer58)). Mutations affecting pre-mRNA splicing account for at least 15% of disease-causing mutations and up to 50% of all mutations described in some genes. Since natural and regulatory splice sites exhibit considerable sequence variation, interpreting the phenotypic effects of these variants on splicing is complex [11].

To detect novel deep intronic mutations and determine their pathogenicity, combining intronic region sequencing with qRT-PCR analysis is essential. Expression was analyzed at the mRNA level since the two variants generate premature stop codons, likely triggering a mechanism of nonsense-mediated decay. Examining the patient’s transcriptome helps rapidly detect abnormal splicing isoforms. Reduced levels of mutant transcripts typically indicate a disease-causing mutation that disrupts normal splicing and targets abnormal mRNAs for degradation, or inactivates a transcriptional regulatory motif [12]. Our results show that both new mutations act as loss-of-function variants, impairing protein translation. Further sequence studies are needed to understand exactly how these mutations alter the coding regions. Notably, our methodologies allowed us to confirm for the first time the pathogenicity of the *ANOS1* and *IL17R* variants, which were previously classified as likely pathogenic according to ACMG criteria.

Although it is crucial to distinguish true causative variants from VUS that may be incorrectly classified as disease-causing, we consider it equally important to demonstrate any potential causal link between these variants and the phenotype when there is strong suggestive evidence of pathology. The widespread use of NGS and the increasing number of candidate genes for CHH have led to the identification of numerous VUS, for which there is insufficient evidence to confirm their involvement in the disease. As such, functional characterization is vital for the accurate interpretation of gene variants and is often necessary to prevent new variants from being misclassified as VUS. These variants often represent incidental findings, and over time, many of them are (re)classified as benign. However, some of them could eventually be identified as causative variants.

In the third case of adult-onset hypogonadism with spontaneous puberty and late-onset ataxia, a missense heterozygous variant in the *WDR11* gene (c.2306T>G [p.Met769Arg]) and a new mutation in the *DCC* gene (c.3533C>T [p.Ser1178Phe]) were identified. *WDR11* missense heterozygous mutations have been reported in CHH patients, some of whom also have mutations in a second known gene [13,14,15]. Similarly, heterozygous DCC mutations have been found in KS and CHH probands [16]. Molecular modelling analysis was performed for the *WDR11* variant because this was the only candidate among those identified in our study for which a reliable three-dimensional structure (AlphaFold) was available at the time of analysis. Importantly, in August 2024, Deng et al. published the cryo-EM structure of the human WDR11-FAM91A1 complex [8]. Our molecular findings suggest that the M769R mutation in *WDR11* affects the structural stability and dynamic behavior of the protein, resulting in notable shifts in flexibility and intradomain motions. These changes, particularly within the “helix zone” and in interactions between key regions, point to mechanisms by which this mutation could disrupt the protein’s function. In this case, we emphasize the relevance of the *WDR11* gene mutation for its role in GnRH neuron migration and normal ciliogenesis, which may account for the failure in fertility restoration following gonadotropin stimulation. Kim et al. (2010) specifically focused on a protein that interacts with *WDR11*, EMX1, a domain transcription factor involved in the development of olfactory neurons [13].

CHH is rare in females, and estrogen-progestin hormone replacement therapy is essential for maintaining bone health, female appearance, improving sexual function, and promoting overall well-being. However, evidence for the restoration of reproductive potential remains limited. The approach to stimulate fertility is not standardized and is believed to be personalized, particularly regarding the type and dosage of gonadotropins used in these patients. Given that both *WDR11* and *DCC* are involved in different brain development processes, we hypothesize that there may be a link between the variants identified and the neurological phenotype observed in this patient.

Currently, *POLR3A*, *POLR3B*, *OTUD4*, *STUB1*, *PNPLA6*, and *RNF216* are all genes associated with CHH syndromes and cerebellar ataxia, with clinical manifestations that can vary and may present with infant or adult onset [17,18,19,20,21,22]. Although further studies are needed to validate our observations and to better understand the exact structural effects of *WDR11* mutation, molecular dynamics simulations provide a promising foundation for future drug discovery and drug repurposing efforts. These efforts could aim at targeting the mutant form of *WDR11* to restore stability and potentially correct the associated dysfunctions.

We report the only female case in our cohort with olfactory bulb anomalies. She carried a new missense heterozygous intronic variant in *FGF8* gene of maternal origin and a de novo heterozygous variant in the *IL17RD* gene (c.1622C>T; p.Ser541Phe), both of which are classified as VUS. Since both *FGF8* and *IL17RD* are implicated in pathogenesis of CHH, we believe the presence of these multiple genetic defects might have synergistically contributed to the CHH phenotype.

In a separate case, a patient with a de novo CHH-associated *DCC* variant presented with severe osteoporosis and reverse hypogonadism at 28 years of age. Hypogonadism is a well-established cause of secondary osteoporosis in men, and the association between low bone mineral density and severe sex steroid deficiency in CHH patients is well recognized. Some studies suggest that bone depletion is most significant when estradiol levels are below 10 pg/mL and/or testosterone levels fall below 6.94 nmol/L [23]. With appropriate long-term treatment, many of the long-term effects of hypogonadism can be minimized. While sex steroid treatment improves bone density, it does not fully reverse the osteoporosis phenotype. Mutations in *FGF8*, *FGFR1*, or *SEMA3A* may not only cause GnRH deficiency but also directly affect bone metabolism [20].

## 4. Materials and Methods

We enrolled 14 patients, in late adolescence and adult age, referred to the Rare Diseases Service, Endocrinology Unit, “Renato Dulbecco” University Hospital, with suspected or diagnosed CHH.

All participants CHH had been diagnosed and classified as KS or nHH according to standard criteria of inclusion:-low serum testosterone concentrations in the setting of low gonadotropin concentrations (<264 ng/dL in adults);-delayed/absent pubertal development, defined by lack of testicular enlargement (volume < 4 mL with Prader orchidometer) in boys at age 14 and absence of breast development in girls at age 13; and/or-infertility/sterility; and/or-criptorchidism and micropenis in the neonatal period; and/or-anosmia/hyposmia assessed by Sniffin’ test (Sense Trading BV, Groningen, The Netherlands), a validated measure of olfactory function; and/or-developmental anomalies, such as cleft lip and palate, dental agenesis, renal agenesis, hypospadias, bimanual synkinesis, sensorineural deafness, and skeletal anomalies; and/or-no identifiable cause of hypothalamic or pituitary dysfunction, such as chronic diseases, pituitary tumors, head trauma; and/or-presence of family cases of CHH and/or infertility sine causa.

CHH reversal was defined as the spontaneous recovery of hypothalamic–pituitary–gonadal axis function off-hormonal treatment in patients with a confirmed diagnosis of CHH, as evidenced by either normalized concentrations of sex steroids and gonadotropins or active spermatogenesis on seminal fluid analysis or fertility off-hormonal treatment. Following treatment-specific washout, all patients with CHH reversal showed typical serum testosterone concentrations with typical LH and FSH concentrations.

### 4.1. Instrumental and Molecular Analyses

After obtaining informed consent, a peripheral venous blood sample was collected (in EDTA). DNA was extracted from the peripheral lymphocytes (extraction KIT from Nuclear Laser Medicine S.r.l., Settala, Italy). and was then quantified using the qubit by an established panel with NGS to search for variants in the putative genes candidates for hypogonadotropic hypogonadism.

The proposed NGS panel was the following: *GNRHR*, *GNRH1*, *KISS1*, *RGPR54*, *KISS1*, *TAC3*, *TACR3*, *FSHB*, *LHB*, *ANOS1*, *FGFR1*, *FGF8*, *PROK2*, *PROKR2*, *CHD7*, *SEMA3A*, *HS6ST1*, *SOX10*, *SEMA7A*, *SEMA3E*, *IL17RD*, *KLB*, *SPRY4*, *DUSP6*, *HESX1*, *FGF17*, *POLR3A*, *POLR3B*, *PNPLA6*, *PLXNA1*, *FLRT3*, *WDR11*, *NELF*, *NTN1*, *DCC*, *FEZF1*, *NROB1*, *NR5A1*, *HESX1*, *LHX4*, *PROP1*, *TUBB3*, *LEPR*, *PCSK1*, *IGSF10*, *FTO*, and *EAP1*.

The variants found were analyzed according to the American College of Medical Genetics and Genomics (ACMG) criteria to define their pathogenicity. To acquire the ACMG criteria important for variant classification, the following steps were followed:

1. The extension of the search for variants found by Sanger sequencing to close relatives (parents, brothers and sisters, etc.) to establish the segregation of the variant found and identify, whether it is de novo or inherited;

2. The control, on international databases (e.g., gnomAD), of the frequency of the variants found in the reference population.

For the estimation of the pathogenicity of variants, we used prediction programs for missense variants (SIFT, polyphen 2, predictSNP, and CADD) for in-frame deletion or insertion variants (PROVEAN, etc.) and for variants that influence splicing (HSF, NetGene2, etc.) with the help of VARSOME.

3. The search for the possible presence of the variants found in international databases of gene variants (ClinVar, LOVD, HGMD, etc.).

One of the so-called “strong” pathogenicity criteria (PS3), present in the ACMG rules for the interpretation of gene variants, is based on the demonstration that the new variant shows its pathogenicity in functional studies. Functional characterization is, therefore, of capital importance for the correct interpretation of gene variants and is often essential to prevent new variants from being classified as VUS. Knowledge of the exact functional meaning of the variant is, in some cases, of extreme relevance for conducting a personalized therapeutic approach. To study gene variants from a functional point of view, different approaches are possible, also depending on the type of variant you want to study (missense mutations, mutations that interfere with the splicing process, and loss-of-function mutations). The approaches, listed below, were followed during this work.

i.Expression of the variant in a biological sample.

The presence of the variant can determine the production of an aberrant or quantitatively insufficient mRNA or protein, which can be identified directly in an appropriate biological sample coming from the subject affected by the genetic pathology. The anomalous mRNA will be highlighted by RT-PCR and subsequent sequencing of the amplified product or analyzed quantitatively by RT-PCR. Aberrant or functionally abnormal or even quantitatively insufficient protein expression has been analyzed by Western blot or immunofluorescence. The easiest biological sample to recover is whole blood. Depending on the genes found mutated in the patients, it will be necessary to ensure that the genes are expressed in the cells of the blood or another biological sample that can be easily collected from the patient. This can be ascertained by consulting gene expression databases (e.g., Human Protein Atlas).

ii.Investigation of the variant through MDs and dynamic cross-correlation analysis.

The presence of the variant can cause effects on the protein, determining structural instability and dynamic behavior alterations with destruction of protein function.

### 4.2. Reverse Transcription and Quantitative Real-Time Amplification (qRT-PCR)

Total RNA was extracted from patients using the QIAmp RNA blood mini kit according to the manufacturer’s protocol; the integrity and purity of the isolated total RNA were assessed using a NanoDrop Spectrophotometer. Complementary DNA (cDNA) was synthesized from 1 µg of total RNA using the cDNA Reverse Transcription Kit (ThermoFisher, Waltham, MA, USA). The qRT-PCR was performed using Power Track SYBR Green Master Mix (Applied Biosystems, Waltham, MA, USA), containing the internal reference (ROX). Each qRT-PCR reaction comprised 10 μL 2× SYBR Green PCR Master Mix, forward and reverse primers at optimized concentrations of 400 nM (final concentration). The sequence of primers was the following: *ANOS1* forward (5′-TTTGTGAGCCTCTCTTCCCC-3′), *ANOS1* reverse (5′-TCCGTAGTCTTTCTCCGCTT-3′), *IL17RD* forward (5′-ACAGCCGGACAATCTAGCTT-3′), and *IL17R* reverse (5′-CAGCTGGTCGTCTCTGTAGT-3′). The thermal profile consisted of enzyme activation at 95 °C for 2 min, followed by 40 cycles of denaturation at 95 °C for 5 s, and a combination of annealing and extension at 60 °C for 30 s. Each qRT-PCR experiment included triplicate no-template controls and patient samples for all primers tested. We used β-actin to normalize the data. Relative expression was calculated using the comparative cross-threshold (ΔΔCt) method as previously described [24].

### 4.3. Molecular Dynamics Simulations and Dynamic Cross-Correlation Analysis

In the absence of an experimentally resolved crystal structure for *WDR11*, we selected the AlphaFold-predicted model (AlphaFoldDB code AF-Q9BZH6) for our computational analysis [25]. To specifically examine the protein domain containing the M769R mutation, the AlphaFold model was truncated to include only residues from Ala465 to Ala828. Thus, the M769R mutant model was generated from the WT structure using UCSF Chimera 1.17.3 software [26]. Methionine at position 769 was replaced with arginine, utilizing the Dunbrack 2010 rotamer library to ensure accurate torsion angles and rotamer probabilities. To stabilize the termini, acetyl (ACE) and N-methyl (NME) caps were manually added to the N- and C-termini, respectively. Topology files for the systems were generated using the tleap program from AmberTools and subsequently converted into GROMACS format with ParmEd. The ff19SB Amber force field was applied to parametrize the protein [27], and all simulations were conducted using GROMACS 2021. The receptor was solvated with the optimal point charge (OPC) water model, with a 12 Å cutoff applied for short-range interactions. Long-range electrostatics were computed with the particle mesh Ewald (PME) method [28] using a grid spacing of 1.2 Å under periodic boundary conditions. The non-iterative LINCS algorithm was applied to constrain bond lengths, enabling the use of a 2 fs integration time step. To resolve steric clashes, each system was subjected to a 20,000-step energy minimization in two steps using the steepest descent algorithm. In the first step, C-alpha atoms of the protein were fixed to allow relaxation of side chains, other atoms, and water molecules. In the second step, all atomic positions were fully minimized. After minimization, each complex was gradually equilibrated and heated to 300 K. The equilibration protocol employed an iterative approach, alternating between isothermal-isobaric (NPT) and canonical (NVT) ensemble simulations to achieve uniform atom distribution and box size stabilization. Starting at 100 K, the system was first simulated at 1 atm in the NPT ensemble for 1 ns, then in the NVT ensemble for 1 ns, allowing the solvent to adjust and preventing low-density regions. Velocities were only generated during the initial NPT step and then carried over between steps. The temperature was then raised by 50 K, repeating the NPT/NVT cycle until reaching 250 K. Subsequently, the temperature increments were reduced to 25 K per cycle to ensure gradual heating as the system approached the water melting point. During equilibration, position restraints were applied to the protein’s C-alpha atoms. An initial force constant of 1000 kJ/mol was applied in the first 50 K NPT/NVT cycle and progressively reduced by 50% with each temperature increment to facilitate smooth protein equilibration. In steps 5 and 6 of equilibration, the force constant was further reduced from 125 kJ/mol to 75 kJ/mol and 25 kJ/mol, respectively, with a low 25 kJ/mol restraint maintained upon reaching 300 K. For the equilibration phase, Berendsen thermostat and barostat were used to avoid abrupt changes in temperature and pressure. Before production simulations, these were switched to the stochastic velocity rescaling and Parrinello–Rahman algorithms, respectively. A 10 ns NPT preproduction run at 300 K was then conducted to eliminate the influence of initial conditions. RMSD analysis was conducted using MDAnalysis 2.4.3, a Python library designed for the analysis of molecular dynamics (MD) trajectories across a wide range of common formats. MDAnalysis also allows for output in various formats, supporting both atom selections and compatibility with visualization or analysis tools. All graphical representations were generated using Matplotlib 3.8.2, a comprehensive Python library for creating static, animated, and interactive visualizations. The dynamic cross-correlation matrix (DCCM) was built by taking into account the coordinates of the C-alpha atoms, enough to describe the largest system motions, using the cpptraj plugin from AmberTools22 [29].

Each element Cij of the cross-correlation (or normalized covariance) matrix is calculated as:

Positive Cij values represent a correlated motion between residues i and j (i.e., the residues move in the same direction). Negative values of Cij represent an anti-correlated motion between residues i and j (i.e., they move in opposite directions).

## 5. Conclusions

The clinical and metabolic characteristics of our cohort are in line with those reported in the literature. In our cohort of CHH/KS patients, we identified two cases with novel, likely pathogenic variants in *ANOS1* and *IL17RD* genes, which have been demonstrated to be pathogenic with functional studies, as well as two novel VUS variants. The other variants classified as VUS are all rare. Through molecular dynamics simulation, we demonstrated that the M769R mutation in *WDR11* impacts the structural stability and dynamic behavior of the protein, potentially disrupting its function. Despite our extensive analysis, two patients remained without a genetic diagnosis, even though they presented with a suggestive phenotype. These mutation-negative patients may have unidentified causative genes not included in our gene panel or more complex genetic alterations that are not easily detected by NGS.

While it is critical to distinguish true causative variants, especially from VUS that may be incorrectly categorized as disease-causing, we believe it is equally important to demonstrate any causal link between these variants and the phenotype, particularly when the phenotype strongly suggests pathology. For this reason, we advocate for genetic functional characterization to investigate new gene variants previously classified as VUS, as well as new pathogenetic variants, especially when the disease phenotype is clear. This should be performed whenever possible in all patients with hypogonadotropic hypogonadism.

## Figures and Tables

**Figure 1 ijms-26-05207-f001:**
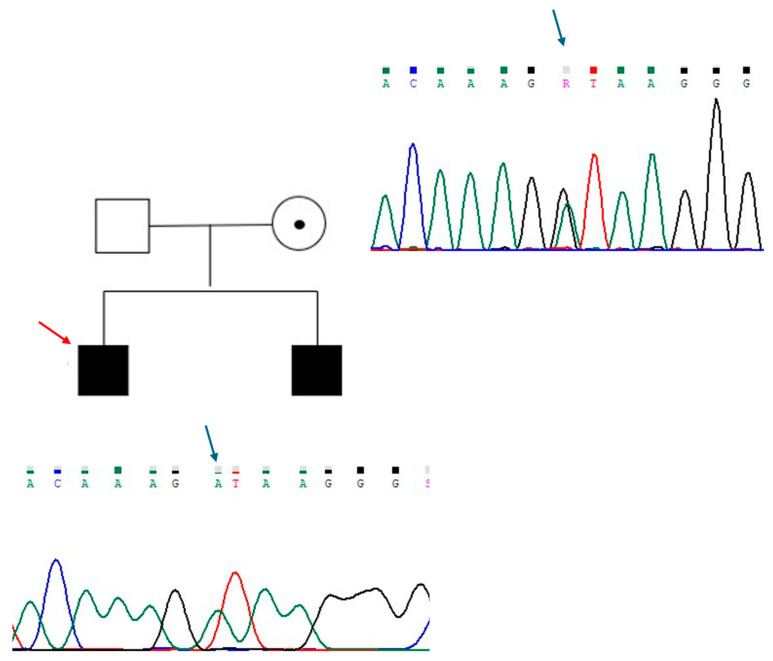
Genealogic tree of patient 1 family and electropherograms showing the region of intron 4 of *ANOS1* containing the variant c.541+1G>A. The substituted base is indicated with a blue arrow. The proband is indicated with a red arrow. Each electropherogram is illustrated next to the respective individual.

**Figure 2 ijms-26-05207-f002:**
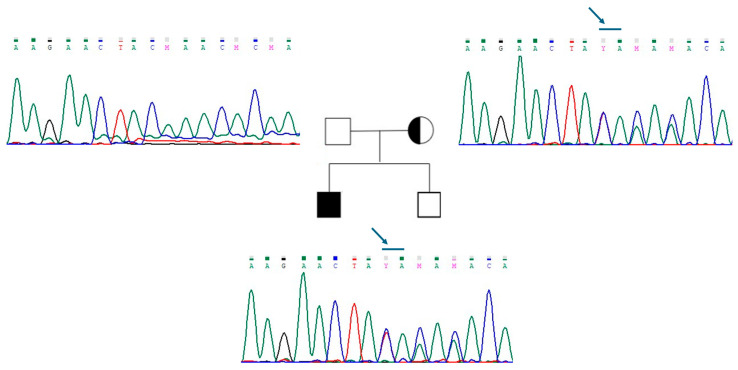
Genealogic tree of patient 2 family and electropherograms of the region of exon 12 of *IL17R*. The variant c.1303_1304dup (p.Lys436ThrfsTer58) is indicated with a blue arrow. The variant is present in the proband and in his mother. Each electropherogram is illustrated next to the respective individual.

**Figure 3 ijms-26-05207-f003:**
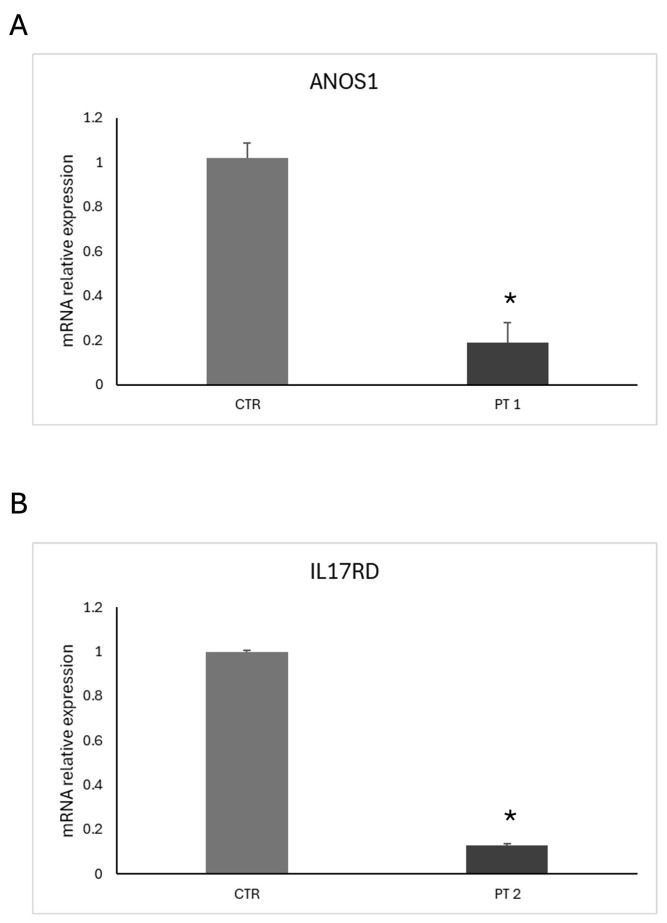
qRT-PCR analysis of *ANOS1* and *IL17RD* mRNA expression levels in patients with CHH. (**A**) Relative *ANOS1* mRNA expression in peripheral blood leukocytes from patient 1 (PT 1) compared to a healthy control (CTR). (**B**) Relative *IL17RD* mRNA expression in peripheral blood leukocytes from patient 2 (PT 2) compared to a healthy control (CTR). Total RNA was extracted from whole blood, and gene expression levels were normalized to β-actin. Data represent the mean ± SEM of three independent experiments performed in triplicate. * *p* < 0.05 vs. control.

**Figure 4 ijms-26-05207-f004:**
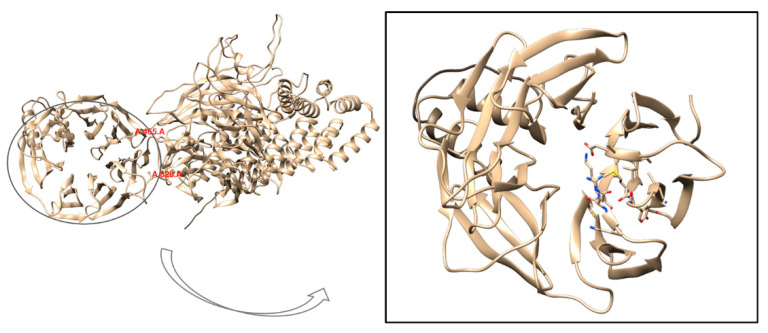
3D structure of the full *WDR11* protein obtained from the AlphaFold model (AlphaFoldDB code AF-Q9BZH6) shown in cartoon representation and faded yellow color. The focus is on the truncated protein domain containing the M769R mutation, with the AlphaFold model truncated to include residues from Ala465 to Ala828.

**Figure 5 ijms-26-05207-f005:**
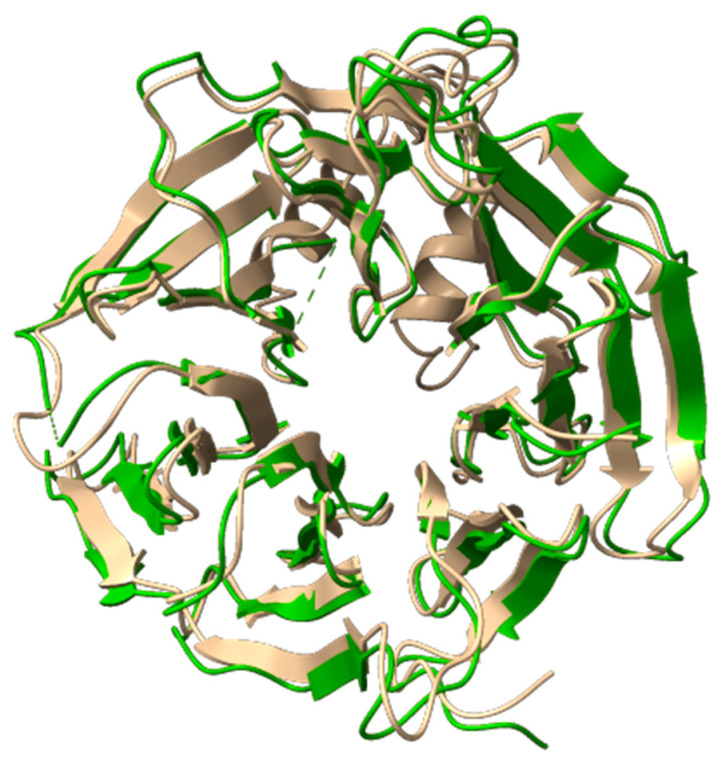
Superimposition of the 3D structure of the WDR11 protein derived from the AlphaFold model (AlphaFoldDB code AF-Q9BZH6) and the cryo-EM model (PDB code 8Z9M). The AlphaFold model is shown in faded yellow (cartoon representation) and the cryo-EM structure in green (cartoon representation).

**Figure 6 ijms-26-05207-f006:**
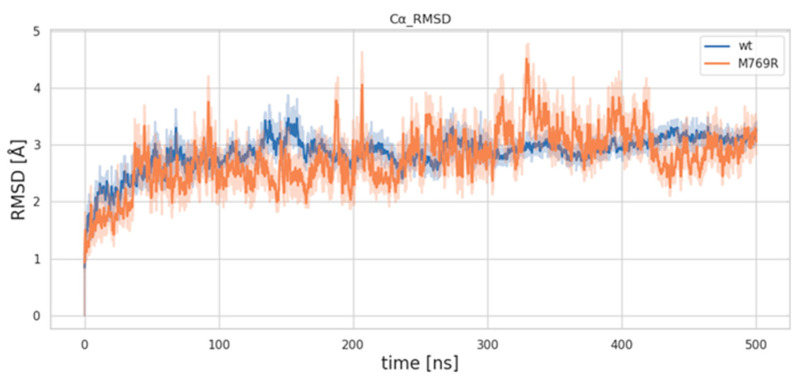
RMSD trends of C-alpha atoms for the wild-type (WT) and M769R mutant structures of WDR11.

**Figure 7 ijms-26-05207-f007:**
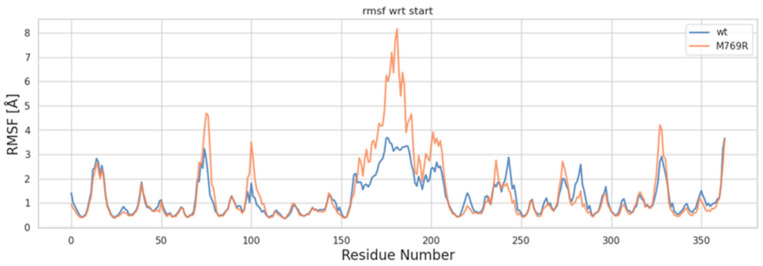
RMSF plot of C-alpha atoms for the wt and M769R mutant structures of WDR11.

**Figure 8 ijms-26-05207-f008:**
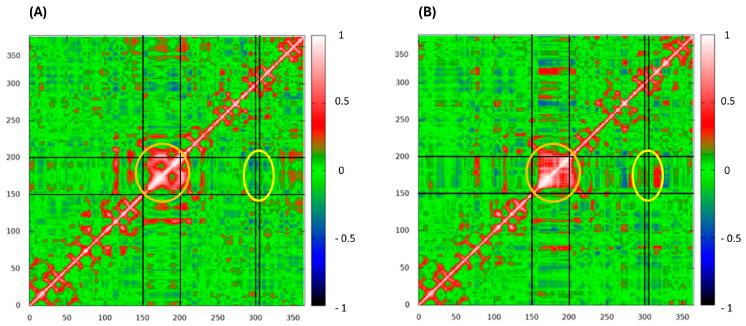
Dynamic cross-correlation matrices (DCCM) of wild-type complex (**A**) and M769R mutant (**B**) complex. Red stands for correlation and blue for anti-correlation. The orange circle highlights the helix zone’s self-correlation (residues 150–200), while the yellow circle indicates the impact of the mutation (residue 306) site’s motions on the helix region.

**Table 1 ijms-26-05207-t001:** Clinical, genetic, and radiological characteristics, hormonal values, and therapy management in CHH/KS patients. AMH: anti-Müllerian hormone; CHH: congenital hypogonadotropic hypogonadism; EP: estroprogestinics; F: female; FSH: follicle-stimulating hormone; KS: Kallmann syndrome; LH: luteinizing hormone; M: male; MRI: magnetic resonance imaging; nHH: normosmic hypogonadotropic hypogonadism; Pt: patient.

	Pt 1	Pt 2	Pt 3	Pt 4	Pt 5	Pt 6	Pt 7	Pt 8	Pt 9	Pt 10	Pt 11	Pt 12	Pt 13	Pt 14
**Age at diagnosis**	52 years	18 years	22 years	18 years	42 years	7 months	18 years	17 years	24 years	16 years	18 years	30 years	17 years	17 years
**Sex**	M	M	F	M	F	M	M	M	M	F	M	F	M	M
**Phenotype**	KS	nHH	nHH	KS	KS	KS	nHH	KS	nHH	KS	KS	nHH	nHH	nHH
**Clinical characteristics**	Hypogonadism Anosmia	Hypogonadism	Hypogonadism Secondary amenorrhea Ataxia	Hypogonadism Anosmia Eunuchoidal proportions	Hypogonadism Primary amenorrhea Anosmia	Cryptorchidism Micropenis Hypogonadism Anosmia	Hypogonadism	Hypogonadism Anosmia Eunuchoidal proportions	Hypogonadism Osteoporosis	Hypogonadism Primary amenorrhea Iposmia	Hypogonadism Gynecomastia Anosmia Eunuchoidal proportions	Hypogonadism Secondary amenorrhea	Cryptorchidism Hypogonadism Eunuchoidal proportions	Hypogonadism Eunuchoidal proportions
**Puberty**	Absent	Absent	Normal	Absent	Absent	Absent	Absent	Absent	Absent	Absent	Absent	Normal	Arrest	Absent
**CHH reversal**	No	No	No	No	No	No	No	No	Yes	No	No	No	No	No
**Gene**	*ANOS1*	*IL17RD*	*WDR11* *DCC*	*PROK2*	*FGF8* *IL17RD*	*-*	*IL17RD*	*PROK2*	*DCC*	*CHD7* *TACR3*	*FGFR11*	*TACR3*	*-*	*DUSP6*
**LH (IU/L)**	0.05	0.16	0.80	0	0.10	0.20	0.38	0.39	0.39	0.40	0.18	0.25	1.58	0.30
**FSH (IU/L)**	1.48	3.05	0.90	0.68	0.34	0.70	1.56	0.54	0.54	2.15	1.30	0.90	1.89	1.05
**Testosterone (ng/mL) or estradiol (pg/mL)**	113	20	10	0.30	10	0.03	0.18	14	14	11.80	14	15.2	19	12
**Inhibin B (pg/mL)**	156	210		175		165	182	200	180		156		223	210
**AMH (ng/mL)**			3		1.2					4.2		2.6		
**Testicular volume (cm^3^) or ovarian volume (cm^3^)**	3.0	2.0	5.0	2.0	2.0	1.0	<2.0	2.0	4.0	<2.0	2.0	2.5	6.0	2.0
**Brain MRI**	Normal	Normal	Cerebellar atrophy	Normal	Hypoplasia of olfactory bulbs	Normal	Normal	Normal	Normal	Hypoplasia of olfactory bulbs	Normal	Normal	Normal	Normal
**Therapy**	Testosterone/Gonadotropin	Gonadotropin/Testosterone	Estrogen/Gonadotropin/EP	Gonadotropin/Testosterone	Estrogen/Gonadotropin/EP	Gonadotropin/Testosterone	Testosterone/Gonadotropin	Gonadotropin	Testosterone	Estrogen	Gonadotropin/Testosterone	EP	Gonadotropin	Testosterone

**Table 2 ijms-26-05207-t002:** Genetic characteristics of CHH/KS variants. AD: autosomal dominant; *ANOS1*: anosmin 1; AR: autosomal recessive; *CHD7*: chromodomain helicase DNA binding protein 7; *DCC*: deleted in colorectal cancer; *DUSP6*: dual specificity phosphatase 6; *FGF8*: fibroblast growth factor 8; *FGFR11*: fibroblast growth factor receptor 1; *IL17RD*: interleukin 17 receptor D; LOF: loss of function; *PROK2*: prokineticin 2; *TACR3*: tachykinin receptor 3; VUS: variant of uncertain significance; *WDR11*: WD repeat domain 11.

Gene	RefSeq/Variant Identified	Location	Consequence	Impact	Genotype	Inheritance	Clinical Significance
*ANOS1*	NM_000216.4: c.541+1G>A GRCh37: ChrX:g.8565074C>T	Intron 4	-	LOF	Hemizygous	AD	Probably pathogenic
*IL17RD*	NM_017563.5: c.1303_1304dup GRCh37: Ch3:g.57132427_5713	Exon 12	(p.Lys436ThrfsTer58)	Frameshift LOF	Heterozygous	AR/AD/Oligo	Probably pathogenic
*DCC*	NM_005215.4 c.3533C>T GRCh37: Chr18:g.50985742C>T	Exon 24	(p.Ser1178Phe)	Missense	Heterozygous	AR/AD/Oligo	VUS
*WDR11*	NM_018117.12 c.2306T>G GRCh37	Exon 18	(p.Met769Arg)	Missense	Heterozygous	AD	VUS
*PROK2*	NM_001126128.2 c.146G>T	Exon 2	(p.Ser49Ile)	Missense	Heterozygous	AR/AD/Oligo	VUS
*FGF8*	NM_033163.5 Chr1o: g.103530393C>T GFCh37	Intron 5	-	Missense	Heterozygous	AD/Oligo	VUS
*IL17RD*	NM_017563.5 Chr3: c.1622C>T	Exon 12	(p.Ser541Phe)	Missense	Heterozygous	AR/AD/Oligo	VUS
*IL17RD*	NM_017563.5 c.692A>G	Exon 7	(p.Tyr231CYS)	Missense	Heterozygous	AR/AD/Oligo	VUS
*PROK2*	NM_001126128.2 c.121G>A	Exon 2	(p.Gly41Ser)	Missense	Heterozygous	AR/AD/Oligo	VUS
*DCC*	NM_005215.4 c.1435A>G	Exon 9	(p.Thr479Ala)	Missense	Heterozygous	AR/AD/Oligo	VUS
*CHD7*	NM_017780.4		(p.Met340Val)	Missense	Heterozygous	AR	Benign
*TACR3*	NM_001059		(p.Lys286Arg)	Missense	Heterozygous		Benign
*FGFR11*	NM_023110.3 c.1058C>G	Exon 8	(p.S353C)	Missense	Heterozygous	AR/AD/Oligo	VUS
*TACR3*	NM_001059.3 Chr4: c.101A>T	Exon 4	(p.Lys361Ter)	LOF	Heterozygous	AR	Probably pathogenic
*TACR3*	NM_001959.3 Chr4: c.824G>A	Exon 3	(p.Trp275Ter)	LOF	Heterozygous	AR	Pathogenic
*DUSP6*	NM_001946.4 c.838+3G>C Chr12:g.89744362C>G hg19/GRCh37	Intron 2	-	Missense	Heterozygous	AR	VUS

## Data Availability

The data presented in this study are available on request from the corresponding author.

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
