# Peer review of "Identification of Novel Genetic Variants in a Cohort of Congenital Hypogonadotropic Hypogonadism: Computational Analysis of Pathogenicity Predictions"

_ijms, 2025, doi:10.3390/ijms26115207_

Round 1
Reviewer 1 Report
Comments and Suggestions for Authors
Congenital hypogonadotropic hypogonadism (CHH) is characterized by absent or incomplete pubertal development due to insufficient production, release, or action of gonadotropin-releasing hormone (GnRH). This deficit results in low levels of luteinizing hormone (LH), follicle-stimulating hormone (FSH), and sex steroids despite otherwise intact pituitary function. CHH occurs in approximately 1 in 4,000 live births, is more commonly diagnosed in males, and exhibits extensive genetic diversity, with over 30 causative genes identified. In this study, a customized panel targeting 46 candidate genes was used to screen individuals with suspected or confirmed CHH/Kallmann syndrome (KS), uncovering a likely genetic explanation in 71 percent of cases. The paper offers rich clinical and genetic data, includes appropriate citations, and holds significant relevance for both patient care and research.
Comments:
“with a phenotypically andgeneticallyapprovalInstitutional Review Board approval heterogeneous disorder like CHH.”?
Table 1 Some clinical characteristic values ​​are missing. Please indicate in the text.
Table 2 The abstract says "using a custom 46 candidate gene panel". There are only a dozen genes listed in the table, so the rest are not mutated? What does "Oligo" mean?
Figure 1 Do both male probands who carry the variant c.541+1G>A. bear mutations in previously reported genes?
Figure 2 Has the woman in the graph been diagnosed with CHH or is showing symptoms of the disease?
Figure 3 A and B are missing from the figure. Why were ANOS1 and IL17R chosen for RT-PCR analysis?
Figure 4 The image is fuzzy and the structural differences between Ala465 and Ala828 are not clear.
Figure 5 The structural changes caused by the CHH mutation are not shown in the figure.
Figure 6 How does the M769R mutation affect the overall structure of WDR11?
Figure 7 How does the M769R mutation affect residues 150-200?
Figure 8 Please mark the differences that readers can follow.
Figure 4-8 lacks experimental validation.
Author Response
Reply to reviewer n. 1
Comments:
C1: “with a phenotypically and genetically approval Institutional Review Board approval heterogeneous disorder like CHH.”?
R1: We thank the reviewer for his/her observation. An Institutional Review Board statement has been included.
C2: Table 1 Some clinical characteristic values ​​are missing. Please indicate in the text.
R2: We thank the reviewer for this comment. Missing values in Table 1 are gender-related, so that we included AMH levels for women and Inhibin B for men.
C3: 3qQQQTable 2 The abstract says "using a custom 46 candidate gene panel". There are only a dozen genes listed in the table, so the rest are not mutated? What does "Oligo" mean?
R3: We thank the reviewer for the question. In Table 2 only mutated genes have been reported. Oligogenic transmission describes a trait influenced by a few genes. Multiple genetic defects might synergize to produce a more severe CHH phenotype. “Oligo” means that it has been recognized in 1.5-15% of cases of CHH.
C4: Figure 1 Do both male probands who carry the variant c.541+1G>A. bear mutations in previously reported genes?
R4: We thank the reviewer for the valuable suggestion. The variants reported are in the ANOS gene.
C5: Figure 2 Has the woman in the graph been diagnosed with CHH or is showing symptoms of the disease?
R5: We thank the reviewer for the comment. The patient has been diagnosed with CHH at the age of 18 because of primary amenorrhoea.
C6: Figure 3 A and B are missing from the figure. Why were ANOS1 and IL17R chosen for RT-PCR analysis?
R6: We thank the Reviewer for pointing this out. The labels “Figure 3 A and B” refer to the qRT-PCR results shown in the corresponding figure panel, where Panel A depicts the expression level of ANOS1 mRNA in patient 1 and Panel B shows the expression level of IL17RD (not IL17R) mRNA in patient 2. The legend has been revised to clarify this, and the figure will be adjusted in the revised manuscript to include proper labeling for panels A and B.
Expression was analyzed at the mRNA level since the two variants generate premature stop codons, likely triggering a mechanism of nonsense-mediated decay (PMID: 32079193).
As for the choice of ANOS1 and IL17RD for qRT-PCR analysis, these two genes were selected because the respective patients (patient 1 and patient 2) harbored novel variants in these genes, which were classified as likely pathogenic according to ACMG criteria. Expression was analyzed at the mRNA level since the two variants generate premature stop codons, likely triggering a mechanism of nonsense-mediated decay (Lambert JM, Ashi MO, Srour N, Delpy L, Saulière J. Mechanisms and Regulation of Nonsense-Mediated mRNA Decay and Nonsense-Associated Altered Splicing in Lymphocytes. Int J Mol Sci. 2020 Feb 17;21(4):1335. doi: 10.3390/ijms21041335). Indeed, for the detection of novel deep intronic mutations and the determination of their pathogenicity, it is essential to complement sequencing data with analysis of mRNA expression. This can be achieved by conventional RT-PCR approaches (Vaz-Drago, R., Custódio, N. & Carmo-Fonseca, M. Deep intronic mutations and human disease. Hum Genet 136, 1093–1111 (2017). https://doi.org/10.1007/s00439-017-1809-4; Choi, W.H., Cho, Y., Cha, J.H. et al. Functional pathogenicity of ESRRB variant of uncertain significance contributes to hearing loss (DFNB35). Sci Rep 14, 21215 (2024). https://doi.org/10.1038/s41598-024-70795-8) The significant downregulation observed for both genes confirmed a likely pathogenic effect.
C7: Figure 4 The image is fuzzy and the structural differences between Ala465 and Ala828 are not clear.
R7: We thank the reviewer for this helpful observation. Figure 4 has been replaced with a higher-resolution version, in which the structural differences between Ala465 and Ala828 are now more clearly visible. In this updated image, it is also easier to appreciate the regions where the original full-length model was trimmed to obtain the portion of interest used in our structural analysis.
C8: Figure 5 The structural changes caused by the CHH mutation are not shown in the figure.
R8: We thank the reviewer for the valuable comment. However, we would like to clarify that the purpose of Figure 5 is not to illustrate structural changes caused by CHH-related mutations, but rather to validate the reliability of the AlphaFold (AF) model used in our computational analysis. As detailed in the text, this figure shows the structural superposition between the AF model and the recently published cryo-EM structure of the WDR11-FAM91A1 complex (PDB: 8Z9M), demonstrating a root-mean-square deviation (RMSD) below 1 Å. This high structural congruence supports the use of the AF model for our subsequent in silico investigations. Therefore, no mutations were applied or analyzed in this specific figure.
C9: Figure 6 How does the M769R mutation affect the overall structure of WDR11?
R9: We thank the reviewer for the insightful question. As shown in Figure 6, the C-alpha atoms the root-mean-square deviation (RMSD) profile of the M769R mutant displays a more irregular and fluctuating trend compared to the wild-type, even beyond the initial equilibration phase (first 100 ns). While both systems remain within a comparable RMSD range (2–4 Å), the mutant exhibits more pronounced peaks (up to 5 Å) and overall less stable behavior throughout the simulation. Since the only difference between the two models is the M769R mutation, we hypothesize that the altered dynamics and reduced structural stability of the mutant could be likely attributable to the mutation itself, possibly reflecting a destabilizing effect on the overall fold of WDR11.
C10: Figure 7 How does the M769R mutation affect residues 150-200?
R10: We thank the reviewer for the question. As shown in Figure 7, the root-mean-square fluctuation (RMSF) analysis suggests an increase in flexibility in the 150–200 residue region of the M769R mutant compared to the wild-type. This segment, which includes two short α-helices predicted by AlphaFold, shows RMSF values exceeding 8 Å in the mutant, indicating more pronounced local fluctuations. This region is inherently flexible and often difficult to resolve in structural studies. Based on our observations, we hypothesize that the M769R mutation may introduce a perturbation in the WD40 domain’s architecture that is indirectly transmitted to this distal region. Since the core of the WD40 domain is composed of highly stable β-sheets, it is plausible that the structural impact of the mutation is not absorbed locally but instead propagates toward a more flexible area such as this, where dynamic effects become more evident.
C11: Figure 8 Please mark the differences that readers can follow.
R11: We thank the Reviewer for the valuable suggestion. In response, we have revised the figure, the corresponding text in the manuscript, and the figure caption to enhance clarity and facilitate interpretation.
Specifically, in Figure 8, we have:
- Added a black box to clearly delineate the regions of interest;
- Included a yellow circle to highlight the correlation between the mutation site (residue 306) and the helix region (residues 150–200), in order to illustrate the effect of the mutation on the dynamic behavior of this region;
- Added an orange circle to emphasize the self-correlation within the helix region (residues 150–200).
We have also updated the figure caption as follows:
“Figure 8. Dynamic cross-correlation matrices (DCCM) of wild-type complex (A) and M769R mutant (B) complex. Red stands for correlation and blue for anticorrelation. The orange circle highlights the helix zone’s self-correlation (residues 150-200), while the yellow circle indicates the impact of the mutation (residue 306) site’s motions on the helix region.”
In addition, we have revised the relevant section of the manuscript text, now reading:
“ To investigate the impact of the M769R mutation on intradomain motions within the protein, Dynamic Cross-Correlation Matrices (DCCM) were calculated for each residue in both configurations (Figure 8). Overall, the mutant M769R displays a shift in correlation patterns compared to the wt. Specifically, the mutant exhibits more extensive anti-correlated regions, indicating an increase in opposing motions between residue pairs. In terms of the helix zone’s self-correlation (residues 150 to 200, orange circle in Figure 8), the wt shows this region as a compact, highly correlated block, indicating coordinated motion within residues 150–200. In contrast, the mutant displayed weaker and more fragmented correlations within this zone, suggesting that the mutation introduces additional flexibility or instability. This observation is consistent with the higher fluctuations observed in the RMSF analysis for this region in the mutant. Regarding the impact of the mutation site’s motions on the helix zone (yellow circle in Figure 8), an anti-correlation is observed between residue 306 and the 150–200 region in the wt protein, indicating opposing motions between these areas. However, in the mutant, this relationship shifts to a positive correlation, suggesting that these regions now move more coordinately. This transition from anti-correlation to correlation indicates that the M769R mutation has altered the dynamic relationship between these regions, likely through new interactions or structural changes that align their motions. Such a shift could lead to a loss of native flexibility or stability, potentially affecting the protein’s overall function. In summary, the M769R mutation appears to induce a more complex and disordered dynamic profile, with expanded regions of both correlation and anti-correlation throughout the structure. This altered dynamic behavior is likely a key factor in changes to protein stability and functionality.”
C12: Figure 4-8 lacks experimental validation.
R12: We sincerely thank the Reviewer for the constructive comment regarding the lack of experimental validation for Figures 4–8. We fully acknowledge the critical importance of experimental validation to support and confirm computational predictions. Given the complexity of this protein, experimental validation is currently challenging. Nevertheless, we recognize its crucial role and plan to address it in future investigations as resources and opportunities allow. As further support, we refer to recent work by Cannarella et al., Int. J. Mol. Sci., 2023, 24(8), 7428, where similar computational approaches are successfully integrated with experimental data to provide mechanistic insights.
Reviewer 2 Report
Comments and Suggestions for Authors
The manuscript is of high interest, but my concern is firstly about the findings on ANOS1 and Il17RD results. In the figures of the electropherograms nucleotide substitutions should be indicated. Moreover the electropherogram in Figure 1 corresponding to the proband has background noise and so is for the one in the second figure. Additionally experiments about the corresponding proteins and their expressions are very limited. How can we be sure about the influence of the mutations on the proetein expression?
Finally, I did not understand why they should have a completely different experimental approach about the WDR11 variant. They should highlight their decision.
I should also propose a conclusion paragraph, including the findings of the study
Author Response
Reply to reviewer 2
C1: The manuscript is of high interest, but my concern is firstly about the findings on ANOS1 and Il17RD results. In the figures of the electropherograms nucleotide substitutions should be indicated. Moreover the electropherogram in Figure 1 corresponding to the proband has background noise and so is for the one in the second figure. Additionally experiments about the corresponding proteins and their expressions are very limited. How can we be sure about the influence of the mutations on the protein expression?
R1: The variant shown in Figure 1 is a base substitution, and in this updated version we have indicated the substituted base. The variant in Figure 2 is a heterozygous duplication (TA bases); the apparent background is due to the different lengths of the two amplified alleles. In this case as well, the duplication site has been indicated. Expression was analyzed at the mRNA level since the two variants generate premature stop codons, likely triggering a mechanism of nonsense-mediated decay (Lambert JM, Ashi MO, Srour N, Delpy L, Saulière J. Mechanisms and Regulation of Nonsense-Mediated mRNA Decay and Nonsense-Associated Altered Splicing in Lymphocytes. Int J Mol Sci. 2020 Feb 17;21(4):1335. doi: 10.3390/ijms21041335).
C2: Finally, I did not understand why they should have a completely different experimental approach about the WDR11 variant. They should highlight their decision.
R2: We thank the reviewer for the insightful question. All other variants in autosomal recessive genes and intronic variants were excluded. Molecular modelling analysis was performed for the WDR11 variant because this was the only candidate among those identified in our study for which a reliable three-dimensional structure (AlphaFold) was available at the time of analysis. Importantly, in August 2024, Deng et al. published the cryo-EM structure of the human WDR11-FAM91A1 complex (PDB IDs: 8Z9M and 8XFB). To assess the quality of computational modelling for WDR11, we compared the AlphaFold model with the available experimental structures, focusing on the WD40-2 domain. The calculated RMSD between AlphaFold and cryo-EM structures was below 1 Å, confirming the high predictive accuracy of the model.
C3: I should also propose a conclusion paragraph, including the findings of the study
R3: We thank the reviewer for his/her proposal; however, such a paragraph is still present at the end of the manuscript after the Materials and Methods section.
Round 2
Reviewer 2 Report
Comments and Suggestions for Authors
As all proposed revisions have been realized, I consider that the paper should be accepted for publication.
Author Response
We thank the reviewer for his/her suggestions.